# Low Dielectric Properties and Transmission Loss of Polyimide/Organically Modified Hollow Silica Nanofiber Composites

**DOI:** 10.3390/polym14204462

**Published:** 2022-10-21

**Authors:** Shu-Yang Lin, Yu-Min Ye, Erh-Ching Chen, Tzong-Ming Wu

**Affiliations:** Department of Materials Science and Engineering, National Chung Hsing University, 250 Kuo Kuang Road, Taichung 402, Taiwan

**Keywords:** hollow silica nanofibers, polyimide, composite, dielectric constant, transmission loss

## Abstract

In this study, a series of low dielectric constant and transmission loss of polyimide (PI)/organically modified hollow silica nanofiber (m-HSNF) nanocomposites were synthesized via two-step polymerization. Two different PIs were fabricated using two types of diamine monomers with or without fluorine-containing groups and biphenylene structure of dianhydride. The chemical structure and morphology of the fabricated composites were characterized using Nuclear magnetic resonance (NMR), Fourier transform infrared (FTIR) and field-emission scanning electron microscopy (FESEM). The two-step polymerization process successfully manufactured and converted from polyamic acid to polyimide after thermal imidization was proved by the NMR and FTIR results. The FESEM and their related energy-dispersive X-ray spectroscopy (EDS) images of nanocomposites indicate that the m-HSNF is extremely dispersed into the polyimide matrix. The high-frequency dielectric constants of the nanocomposite materials decrease as the presence of fluorine-containing groups in diamine monomers and the loadings of the m-HSNF increase. These findings are probably attributed to the presence of the steric hindrance effect brought by trifluoromethyl groups, and the m-HSNF can disrupt the chain packing and increase the free volume, thus reducing the dielectric properties of polyimides. The transmission loss and its related uncertainty of fabricated composite materials contain excellent performance, suggesting that the fabricated materials could be used as substrate materials for 5G printed circuit board.

## 1. Introduction

Recently, polyimide (PI), because of its excellent dielectric and mechanical properties, dimensional stability, and heat resistance, has been extensively applied as electronic packaging materials in microelectronics and substrate materials in flexible printed circuit board (FPCB) [1]. Furthermore, because of the arrival of the 5th generation mobile networks (5G) era, the application frequency and the lead density on the PCB continue to increase with increasing the rapid development of communication technology and the microelectronics industry. These developments evidently cause problems such as cross-talk noise, resistance capacitance (RC) delay, and power loss between the interconnected capacitors. Consequently, the research approach of designed materials with low dielectric constant is increasingly required to meet the high integration degree of the circuit board and high application frequency [2,3,4]. However, the dielectric constant of normal PI is just in the range from 3.1~3.5 [5], which is apparently incapable of meeting their requirements for highly integrated electronic components and requires a lower dielectric constant of substrate materials [6]. In order to reduce the dielectric constant from normal PI, different approaches have been applied and reported by the synthesis of various chemical structures [5,7,8,9] and the organic–inorganic composites [2,10,11,12,13]. For example, Kuo et al. [7] synthesized a series of PI with exotic functional groups including ether, fluorine, ester, amide, ketone, sulfide, sulfone, and alkane groups and proposed that the highly correlated relationship between the dielectric constant of ether and fluorine-based PIs to the fluorine content. Most of these measurements are not the 5G high-band frequency. Min et al. [11] used the template method to synthesize mesoporous silica and incorporated it into PI to prepare a composite containing a low dielectric constant (low-*k*) of 2.6 at 1 GHz. Among these methodologies, the incorporation of porous inorganic material in the PI matrix is a useful method since the low-*k* air (*k* = 1) can be utilized to reduce the dielectric constant of the composites [2,10,11,12,13]. Electrospinning is an effective method to prepare one-dimensional nanotube or nanofiber [14,15], in which numerous air is produced to effectively fabricate low-*k* materials. Moreover, electrospinning techniques compared with normal template strategies to produce nanofibers not only can reduce impurities in the resulting tubular nanomaterials, but also show high efficiencies and low-cost pros [15,16,17]. One-dimensional hollow silica nanofiber (HSNF) fabricated using the electrospinning technique has high mechanical properties, excellent thermostability and lots of porosity, which are beneficial to prepare the low dielectric materials. In order to understand the impact of HSNF on the properties of the composites and to investigate it in detail, the synthesis of PI/organically modified hollow silica nanofiber (m-HSNF) composite using a lot of low-*k* air into the m-HSNF was performed. The more functionalized groups and higher aspect ratio of the m-HSNF proved to be tremendously restraining compared to the more polar groups of polyimides, which produce more air voids and lower dielectric constant.

Recently, the application frequency of the circuits was increased to respond accordingly to the increasing requirements of transmission data. However, the integrity of high-frequency signals may be compromised as a result. In high-frequency circuit transmission, there are two factors responsible for the transmission loss, such as the conductor loss in the metal material circuits and the dielectric loss in the insulating materials [18,19]. In order to investigate the effect of m-HSNF on the circuit transmission loss of the PI/m-HSNF nanocomposites, the micro-strip lines fabricated on a series of PI/m-HSNF nanocomposites were used to measure their circuit transmission loss in the 5G high-band frequency.

In this study, two different polyimides were synthesized using two types of diamine monomers with or without fluorine-containing groups and biphenylene structure of dianhydride. Then a series of PI/organically modified hollow silica nanofiber (m-HSNF) nanocomposites were successfully synthesized by in situ polymerization. The usefulness of the PI/m-HSNF nanocomposites as substrate materials used as 5G FPCB was also considered. Therefore, the dielectric constant of the PI/m-HSNF nanocomposites in the 5G high-band frequency was first measured and reported. Furthermore, we prepared micro-strip lines fabricated on the PI/m-HSNF nanocomposites and measured their transmission loss of the copper circuits in the frequency range from 0.5 GHz~40 GHz. Moreover, the mechanical, thermal, and dimensional stability properties of the PI nanocomposites were studied systematically.

## 2. Experimental

### 2.1. Materials

Dimethyl acetamide (DMAc, C_4_H_9_NO, >99%) and tetraethyl orthosilicate (TEOS, SiC_8_H_20_O_4_, >99%) were purchased from Sigma-Aldrich Chemical Company (St. Louis, MO, USA). Polyvinylpyrrolidone [PVP, (C_6_H_9_NO)_n_] was obtained from JT-Baker Chemical Company. Ammonium hydroxide and (3-aminopropyl)triethoxysilane (APTEOS, C_9_H_23_NO_3_Si, >99%) were obtained from Alfa Aesar Chemical Company (Ward Hill, MA, USA). 2,2-bis [4-(4-aminophenoxy)phenyl] propane (BAPP, C_27_H_26_N_2_O_2_, >98%) was purchased from Tokyo Chemical Industry. Moreover, 3,3′,4,4′-biphenyltetracarboxylic dianhydride (BPDA, C_16_H_6_O_6_, >98%) and 2,2-bis [4-(4-aminophenoxy phenyl)] hexafluoropropane (HFBAPP, C_27_H_20_F_6_N_2_O_2_, >98%) were obtained from Matrix Scientific Chemical Company. BPDA was purified at 165 °C for 24 h under vacuum. DMAc were purified by molecular sieves to remove the water. Other chemicals were utilized as received.

### 2.2. Synthesis of PI/m-HSNF Composites

The hollow silica nanofibers (HSNFs) were fabricated, modified, and reported previously [17]. In brief, the solution of TEOS and PVP was filled into a syringe for electrospinning using a 15-kV high-voltage power. The fabricated HSNF surfaces were functionalized by C_9_H_23_NO_3_Si (APTEOS) coupling agent by dispersing HSNF in ethanol and then APTEOS dispersed in ammonium hydroxide was added in HSNF solution in a three-necked flask while vigorously stirred at room temperature under ultra-sonication for 32 h to obtain APTEOS-modified HSNF (m-HSNF). Two different polyimides prepared by two types of diamine monomers with or without fluorine-containing groups and biphenylene structure of dianhydride were synthesized using two-step polymerization. In brief, the BAPP or HFBAPP dissolved in DMAc was added into a three-necked round-bottomed flask and mechanically stirred under a nitrogen environment. After the mixture was completely dissolved, the BPDA with the same mole of BAPP or HFBAPP was added and stirred for 12 h under the condition of ice-water bath to form a homogeneous polyamic acid (PAA) of BPDA-BAPP (C_43_H_32_N_2_O_8_) or BPDA-HFBAPP (C_43_H_26_F_6_N_2_O_8_). The obtained PAA solution was poured onto the clean glass plate and heated in an oven at 80 °C to eliminate the solvent. Then the thermal imidization of PAA was performed at 100 °C/1 h, 150° C/1 h, 200 °C/1 h, 250 °C/1 h, 300 °C/1 h in vacuum to form polyimide (PI) of BPDA-BAPP (C_43_H_30_N_2_O_7_) or BPDA-HFBAPP (C_43_H_24_F_6_N_2_O_7_). The PI composite films were prepared according to the same procedure as described above.

### 2.3. Characterization and Instruments

The molecular structures and morphology of the prepared BPDA-BAPP/m-HSNF and BPDA-HFBAPP/m-HSNF composites were characterized using nuclear magnetic resonance (NMR), Fourier transform infrared (FTIR), field-emission scanning electron microscopy (FESEM) and transmission electron microscopy (TEM). ^1^H-NMR spectrum was determined by Agilet Technologies DD2 600 MHz 1H-NMR spectroscopy (Santa Clara, CA, USA) using CDCl_3_ as a solvent standard. FTIR spectra obtained in a range from 400–4000 cm^−1^ were operated using a Perkin–Elmer Spectrum One spectrometer (Waltham, MA, USA). The FESEM and TEM measurements were carried out operated using a JEOL JSM-6700F field-emission instrument and JEOL JEM-2010 (JEOL Ltd., Tokyo, Japan). The number-average molecular weight (*Mn*), weight-average molecular weight (*Mw*), and polydispersity PDI = *Mw*/*Mn* of fabricated materials were measured using the Hitachi 2400 gel permeation chromatography (GPC; Waters 717 Plusautosampler, Waters Instruments, Rochester, NY, USA). Narrow molecular-weight distribution of polystyrene standards was used as calibration. The thermal behaviors were determined using TGA 2950 thermal gravimetric analyzer (TA Instruments, New Castle, DE, USA). These experiments were carried out from room temperature to 800 °C at a heating rate of 10 °C/min under a nitrogen environment. The mechanical properties of the resulting polymers and composite materials were carried out on a dynamic mechanical analyzer (DMA8000, Perkin–Elmer, Waltham, MA, USA) from 50 to 275 °C at 2 °C/min heating rate and 1 Hz constant frequency. The tensile strength and elongation at break were measured according to ASTM D882 using the Universal Testing Machine Shimadzu AGS-X (Shimadzu Corporation, Nakagyo-Ku, Kyoto, Japan). The average values of experimental data shown here are obtained from at least three measurements. For the measurement of dielectric properties, the fabricated samples were cut into 10 mm × 10 mm. The dielectric constant was tested by the Anritsu Shockline MS46122B Vector Network Analyzers (Anritsu Corporation, Atsugi, Kanaagawa Prefecture, Japan) from 22 to 41 GHz at room temperature. The high-frequency transmission loss properties of the nanocomposites were measured at room temperature according to Test Methods Manual IPC-TM 650 2.5.5.14 using the Keysight N5247A Microwave Network Analyzer (Keysight Technologies, Santa Rosa, CA, USA) from 0.5 to 40 GHz. These measurements were operated by Taiwan Semiconductor Research Institute, National Applied Research Laboratories (Taipei, Taiwan).

## 3. Results and Discussion

### 3.1. Structural and Morphological Characterizations of Various PI/m-HSNF Nanocomposites 

Two synthesized polyamic acids, BPDA-BAPP and BPDA-HFBAPP, were determined via ^1^H-NMR spectroscopy. Figure 1a shows the ^1^H-NMR data of the BPDA-BAPP and BPDA-HFBAPP. As shown in this figure, the main chain proton signals for amides, aromatics proton of dianhydride, aromatic proton of diamine appeared around 6.5–8.5 ppm. The signals at around 13.0~13.5 ppm and 10.5 ppm correspond to the hydrogen proton of the carboxylic group for the polyamic acid unit and the hydrogen proton of amide after the ring opening process, respectively [20,21]. These results demonstrate that two different polyamic acids using two types of diamine monomers with or without fluorine-containing groups and biphenylene structure of dianhydride were successfully synthesized. The molecular weights of two synthesized polyamic acids determined using GPC are also given in Table 1. The number-average molecular weight (*M_n_*) and polydisperse indices (PDI) of synthesized polyamic acids are in the range from 167,800~169,000 g/mol and 1.24~1.31.

The FTIR methods were used to characterize the chemical structure of the synthesized PI. Figure 1b shows the FTIR spectra in ATR mode of the synthesized BPDA-BAPP films before and after thermal imidization. For the PAA of BPDA-BAPP, the absorption peak for the C-C stretching vibration of the benzene ring obviously appeared at 1503 cm^−1^. The characteristic absorption peak observed at 1650 cm^−1^ is attributed to C=O in the CO–NH. For the BPDA-BAPP PI films, the absorption peaks of the asymmetric and symmetric C=O stretching vibrations in the imide ring were observed at 1780 cm^−1^ and 1721 cm^−1^. The absorption peaks at 1370 cm^−1^ and 722 cm^−1^ were assigned to the C–N stretching vibration and C=O bending vibration in the imide ring. The absence of the characteristic absorption bands at 1650 cm^−1^ implies the complete imidization in the present work. The FTIR spectra of BPDA-HFBAPP are similar to those of BPDA-BAPP, except three more absorption peaks of the C-F stretching vibrations appeared at 1240, 1200 and 1170 cm^−1^. These findings show that the PAA was successfully synthesized and converted into PI using the two-step polymerization process.

The morphology of the fabricated polyimide (PI) coated m-HSNFs synthesized using two-step process was identified using the FESEM and TEM methods. The FESEM and TEM images of 5 wt% BPDA-BAPP/m-HSNF composites are shown in Figure 2a. Both images indicate that the surface of the composite is very smooth and the m-HSNF is extremely dispersed into the BPDA-BAPP polymer matrix. The EDS images of carbon, oxygen and silicon of 5 wt% BPDA-BAPP/m-HSNF nanocomposites are presented in Figure 2b. From these results, the elements of carbon, oxygen, and silicon are well distributed in the composite samples. Similar findings are also observed for BPDA-HFBAPP/m-HSNF nanocomposites.

### 3.2. Physical Properties of Various PI/m-HSNF Nanocomposites 

The thermal behavior of both PI//m-HSNF nanocomposites was examined by TGA analysis. Figure 3 shows the TGA curves of the BPDA-BAPP/m-HSNF nanocomposites. The degradation temperature of 5 wt% weight loss for the neat BPDA-BAPP is slightly lower than those of the BPDA-BAPP/m-HSNF nanocomposites. However, the degradation temperatures of the BPDA-BAPP/m-HSNF nanocomposites are increased with increasing the loading of the m-HSNF. This finding is assigned to the incorporation of the higher thermal stability m-HSNF in the polyimide matrix. Therefore, the thermal stability of the composites increased as compared to the pure polymer matrix. Detailed degradation temperature of 5 wt% weight loss for the BPDA-BAPP/m-HSNF nanocomposites is listed in Table 2. The TGA profiles for the BPDA-HFBAPP/m-HSNF nanocomposites reveals a similar tendency; the degradation temperatures of 5 wt% weight loss obtained from these profiles are summarized in Table 2. Both results reveal that the thermal stability of the PI/m-HSNF nanocomposites is better than those of the pure PI polymer matrices. Similar findings have been reported for related polymer/inorganic material composite systems [22,23].

The storage modulus *E′* against temperature of the BPDA-BAPP/m-HSNF nanocomposites in a temperature ranging from 50 to 275 °C is presented in Figure 4a. These findings reveal that the *E′* of BPDA-BAPP at 50 °C is about 0.97 GPa and decreases as the temperature increases. This result recommends that the BPDA-BAPP molecular motion is limited at the temperature below the glass transition temperature (Tg). As the temperature higher than the Tg, the potential energy barriers of the polymer chain motions are comparable to the thermal energy. The *E′* of the BPDA-BAPP/m-HSNF nanocomposites at 50 °C is increased with increasing the m-HSNF content. Similar results are also obtained for the BPDA-HFBAPP/m-HSNF nanocomposites. Table 2 shows the detailed *E′* at 50 °C for all samples. The enhancement of *E′* may be ascribed to the incorporation of the rigid and stiff m-HSNF, causing the improvement on the rigidity of the polyimide matrix. The glass transition temperatures (Tgs) of the BPDA-BAPP/m-HSNF nanocomposites also determined by DMA test are shown in Figure 4b. These results indicate that the Tg increased with increasing the m-HSNF content. Similar results are also obtained for the BPDA-HFBAPP/m-HSNF nanocomposites. Detailed Tg for all nanocomposites is also illustrated in Table 2. These results suggest that the incorporation of stiff m-HSNF may inhibit the polymer chain motion, causing the increase in glass transition temperatures. Figure 4c presents the curves of the dimensional change versus the temperature of the BPDA-BAPP/m-HSNF nanocomposites. The calculated coefficient of thermal expansion (CTE) values can be obtained from the slope of the curves. The CTE of the prepared BPDA-BAPP/m-HSNF nanocomposites gradually decreased with increasing the m-HSNF content. The CTE was 59.2 ppm/°C for pure BPDA-BAPP and declined to 38.8 ppm/°C for 5 wt% BPDA-BAPP/m-HSNF nanocomposites, revealing the CTE of the nanocomposites was relatively close to that of copper. Similar findings are also observed for the BPDA-HFBAPP/m-HSNF nanocomposites. Detailed CTE for all nanocomposites is also illustrated in Table 2. Normally the substrate materials in the flexible printed circuit boards were coated by copper. The CTE of polyimide and copper is relatively close, which can reduce the delamination in layered PI-Cu composites for microelectronic applications. Due to the presence of m-HSNF, there is much ridge structure of the Si–O–Si groups in the samples. Therefore, the CTE of the prepared nanocomposites is significantly decreased. Nanocomposites with 5 wt% m-HSNF loading revealed the smallest CTE because of the highest amount of Si–O–Si groups of m-HSNF.

The tensile strength and elongation at break of the BPDA-BAPP/m-HSNF nanocomposites are represented in Figure 5. This experimental result shows that the tensile strength of the nanocomposites was increased with the incorporation of m-HSNF. The improvement of tensile strength is attributed to the incorporation of the inorganic and stiff m-HSNF. These results indicate the reinforcement effect of m-HSNF to improve the rigidity of the BPDA-BAPP. The tensile strength of the composite was significantly decreased with increasing the addition of m-HSNF into 5 wt%. This result is probably due to the aggregation effect of the high amount of m-HSNF. In the meantime, the elongation at break of the nanocomposite was decreased with increasing the loading of m-HSNF. This occurrence is ascribed to the reinforcement effect of stiff m-HSNF, resulting in the reduced chain flexibility. The tensile strength and elongation at break of the BPDA-HFBAPP/m-HSNF nanocomposites also show similar tendency. Table 3 presents the detailed tensile strength and elongation at break for all composites.

Figure 6a,b reveals the dielectric properties of the BPDA-BAPP/m-HSNF and BPDA-HFBAPP/m-HSNF nanocomposites versus a frequency in the range from 22~41 GHz. For pure BPDA-BAPP, the dielectric constants at 28 GHz and 38 GHz are about 3.11 and 3.09. The dielectric constant of the BPDA-BAPP/m-HSNF nanocomposites significantly decreased with increasing the amounts of m-HSNF. This result revealed that the m-HSNF containing a certain amount of low-*k* air might play as the typical part of low dielectric fillers and result in the significant decrease in dielectric constants for the 5 wt% BPDA-BAPP/m-HSNF nanocomposites to about 2.79 and 2.76 at 28 GHz and 38 GHz, respectively [24]. It is clear that the dielectric constant of BPDA-HFBAPP is relatively lower than that of BPDA-BAPP. This result may be attributed to the presence of fluorine-containing groups in BPDA-HFBAPP. The dielectric constant of the BPDA-HFBAPP/m-HSNF nanocomposites gradually decreased as the loading of m-HSNF increased. The lowest dielectric constant of 5 wt% BPDA-HFBAPP/m-HSNF nanocomposites can reach about 2.59 and 2.58 at 28 GHz and 38 GHz, respectively. Detailed dielectric constants for all composites is reported in Table 4. This result is probably attributed to the increase in free volume with the presence of m-HSNF. When current went through the polyimide nanocomposites via the dielectric test, two different kinds of substance existed, such as polyimide matrix and m-HSNF. This result is due to the collective effect between the polyimide matrix and m-HSNF. Since water has a relatively high dielectric constant, the data of contact angle and water absorption for the polyimide/m-HSNF nanocomposites are also listed in Appendix A. It is clear that the contact angle of composites was slightly increased with the addition of m-HSNF, but the water absorption was almost the same with the addition of m-HSNF. Therefore, the effect of water absorption on the dielectric constant of fabricated polyimide/m-HSNF nanocomposites can be ignored. A comparison of the dielectric properties in the 5G high-band frequency of BPDA-BAPP/m-HSNF and BPDA-HFBAPP/m-HSNF nanocomposites and formerly described materials were recorded in Table 5. It is evident that both fabricated nanocomposites disclosed better performance of dielectric properties than the formerly described materials.

According to the Test Methods Manual IPC-TM 650 2.5.5.14, the transmission loss properties of the BPDA-BAPP/m-HSNF nanocomposites were measured by high-frequency Vector Network Analyzers and the results are shown in Figure 7a. While transmission loss measurement is completed, it needs to be calibrated according to the test standard, and without proper calibration of the data, the test might easily fail to state the true transmission loss performance of the measurement. After calibration, it needs to find out the closeness of fit in the immediate vicinity of the measurement point in order to estimate the uncertainty at that point in the tested transmission loss. Based on test standard analysis, an uncertainty lower than 15% is suggested to contain high confidence on the tested transmission loss result [27]. For pure BPDA-BAPP, the transmission loss at 28 GHz and 38 GHz were about −2.32 and −2.80 dB/inch. According to the previous literature, when the dielectric constant of the insulating material is lower, its transmission loss is also lower [19,28]. Therefore, the results of the BPDA-BAPP/m-HSNF nanocomposites transmission loss were about −2.09, and −2.40 dB/inch at 28 GHz and 38 GHz, respectively. Similar results of the transmission loss also obtained for the BPDA-HFBAPP/m-HSNF nanocomposites system are shown in Figure 7b. It is clear that the transmission loss of BPDA-HFBAPP is relatively lower than that of BPDA-BAPP. This result may be attributed to the presence of fluorine-containing groups in BPDA-HFBAPP. The transmission loss of BPDA-HFBAPP/m-HSNF nanocomposites gradually decreased as the loading of 5 wt% m-HSNF. Detailed transmission loss for all composites is also illustrated in Table 6. From the data in Table 5, it can be known that the uncertainty of the whole system is less than 15%, indicating that the calibration is successful. It is clear that the transmission loss and its related uncertainty of fabricated materials contain excellent performance. This result suggests that the fabricated materials could be used as substrate materials for 5G printed circuit board.

## 4. Conclusions and Future Perspectives

Organically modified hollow silica nanofibers (m-HSNF) serving as a core were used to fabricate the polyimide/HSNF nanocomposites using two-step polymerization. The two-step polymerization process successfully manufactured and converted from polyamic acid to polyimide after thermal imidization was proved by the NMR and FTIR results. The FESEM and their related energy-dispersive X-ray spectroscopy (EDS) images of nanocomposites indicate that the m-HSNF is extremely dispersed into the polyimide matrix. The CTE of the prepared PI/m-HSNF nanocomposites declined progressively as the m-HSNF content increased. The high-frequency dielectric constants of BPDA-HFBAPP is relatively lower than that of BPDA-BAPP. This result may be attributed to the presence of fluorine-containing groups in BPDA-HFBAPP. The dielectric constants of the nanocomposite materials measured in the range from 22 GHz to 41 GHz decrease as the loadings of m-HSNF increase. This result is probably attributed to the increase in free volume with the presence of m-HSNF. The lowest dielectric constant of 5 wt% BPDA-HFBAPP/m-HSNF nanocomposites can reach about 2.59 and 2.58 at 28 GHz and 38 GHz, respectively. Both results in 5G high-band frequency are significantly lower as compared to the pure BPDA-HFBAPP polymer matrix. The transmission loss and its related uncertainty of fabricated composite materials contain excellent performance, suggesting that the fabricated materials could be used as substrate materials for 5G printed circuit board. The coating of copper foil on the surface of the PI/m-HSNF nanocomposites can be further investigated and simulated in high-frequency ranges.

## Figures and Tables

**Figure 1 polymers-14-04462-f001:**
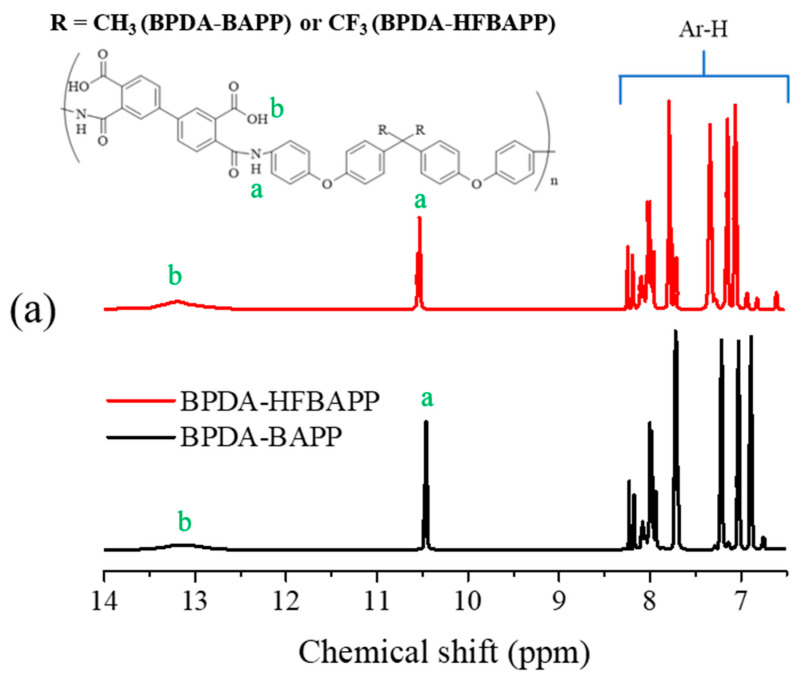
(**a**) ^1^H-NMR spectra of the BPDA-BAPP and BPDA-HFBAPP. The a, b in this figure corresponds to the hydrogen proton of the carboxylic group and the hydrogen proton of amide, (**b**) FTIR spectra of fabricated BPDA-BAPP films before and after thermal imidization.

**Figure 2 polymers-14-04462-f002:**
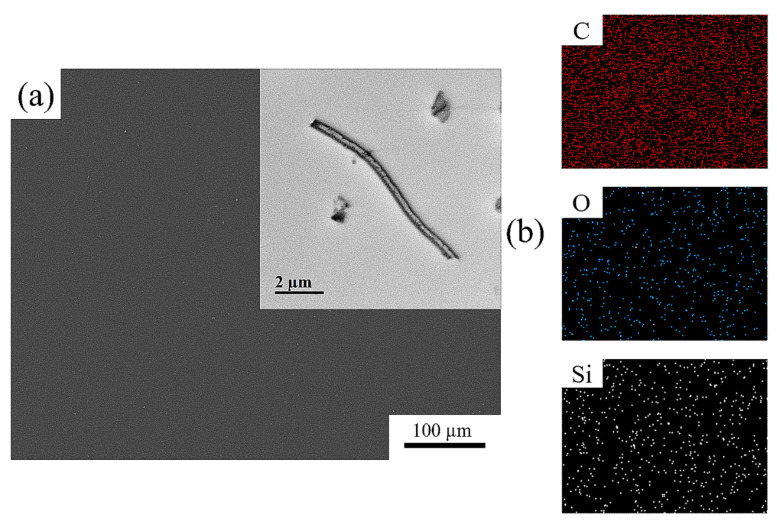
(**a**) FESEM and TEM (insert) images and (**b**) EDS images of carbon, oxygen and silicon of 5 wt% BPDA-BAPP/m-HSNF nanocomposites.

**Figure 3 polymers-14-04462-f003:**
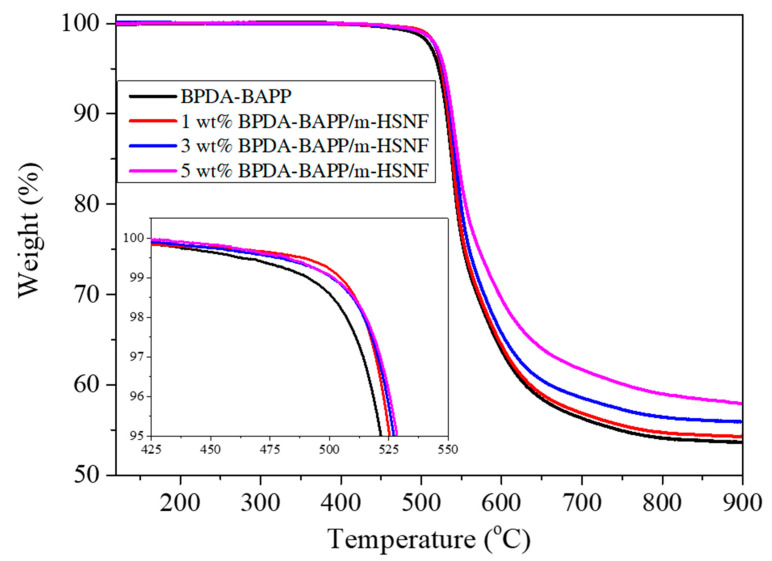
TGA and the temperature of 5 wt% weight loss (insert) curves of BPDA-BAPP/m-HSNF nanocomposites.

**Figure 4 polymers-14-04462-f004:**
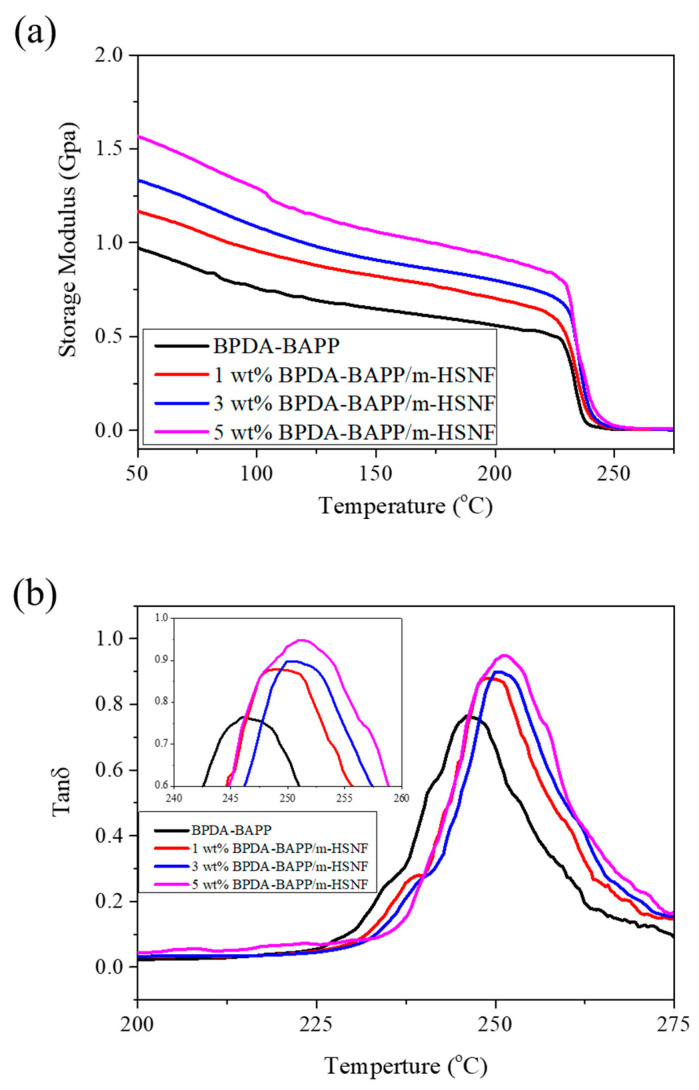
(**a**) Storage modulus *E′*, (**b**) tan δ, and (**c**) dimensional change against temperature of BPDA-BAPP/m-HSNF nanocomposites.

**Figure 5 polymers-14-04462-f005:**
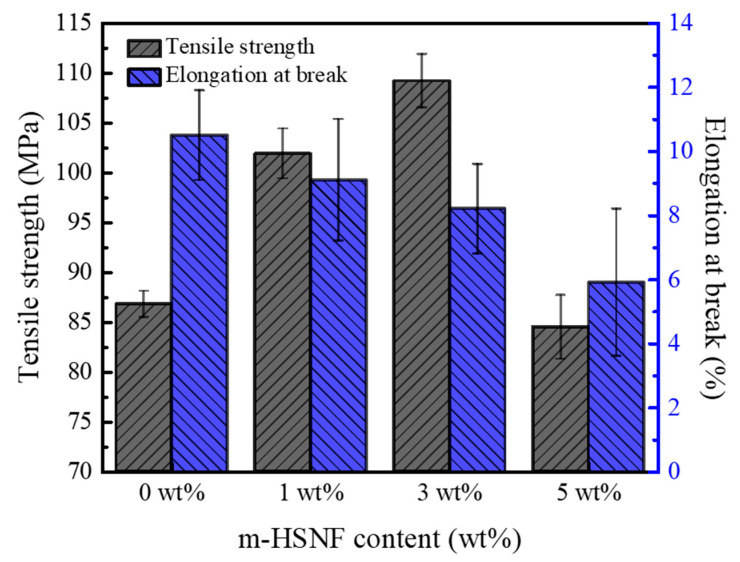
Tensile strength and elongation at break of BPDA-BAPP/m-HSNF nanocomposites.

**Figure 6 polymers-14-04462-f006:**
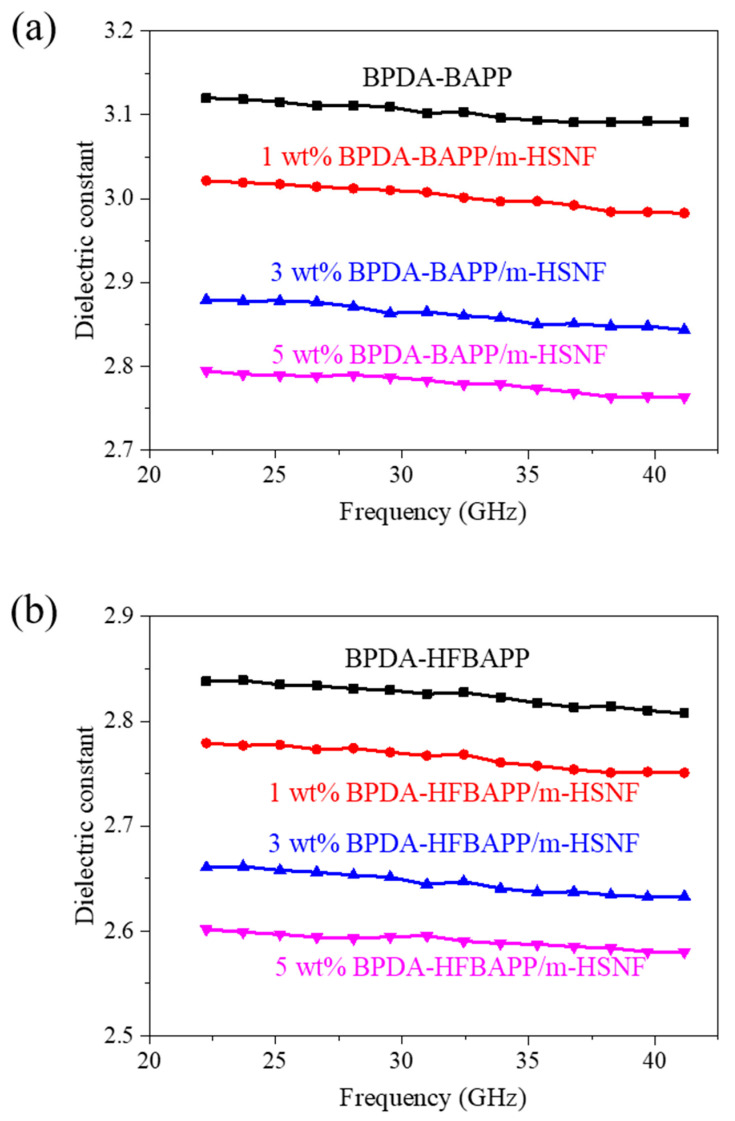
Dielectric properties of (**a**) BPDA-BAPP/m-HSNF and (**b**) BPDA-HFBAPP/m-HSNF nanocomposites.

**Figure 7 polymers-14-04462-f007:**
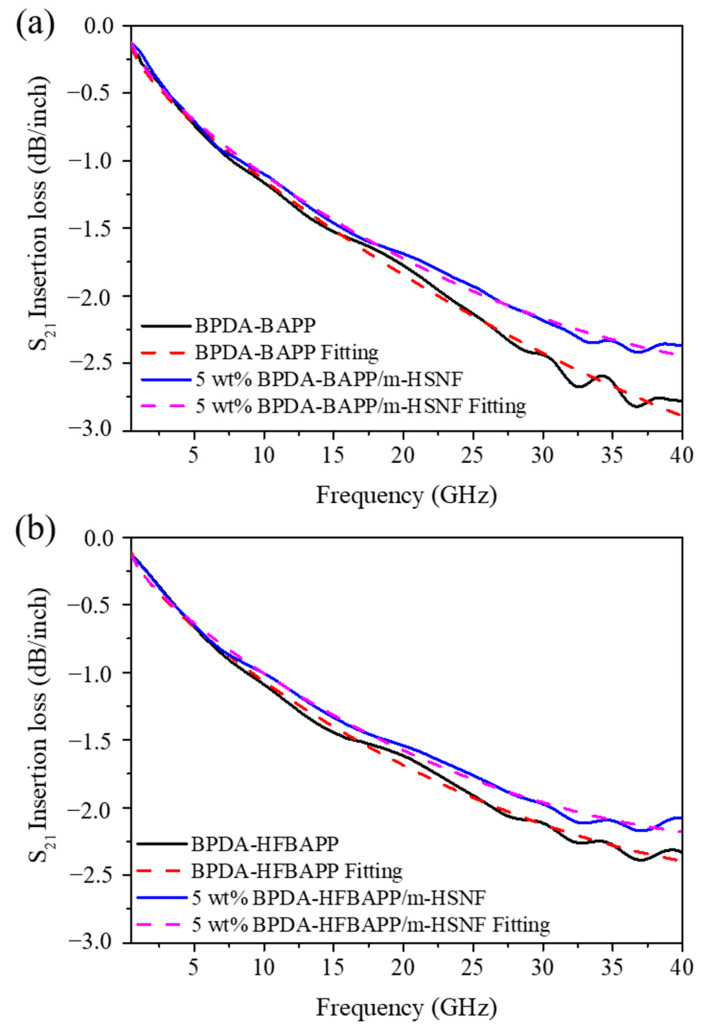
S_21_ insertion loss versus frequency of (**a**) BPDA-BAPP and 5 wt% BPDA-BAPP/m-HSNF nanocomposites and (**b**) BPDA-HFBAPP and 5 wt% BPDA-HFBAPP/m-HSNF nanocomposites.

**Table 1 polymers-14-04462-t001:** Molecular weight and PDI of synthesized polyamic acid.

Polymer	*M_w_* (g/mol) × 10^4^	*M_n_* (g/mol) × 10^4^	PDI
BPDA-BAPP	21.95	16.78	1.307
BPDA-HFBAPP	20.87	16.90	1.235

**Table 2 polymers-14-04462-t002:** Thermal and mechanical properties of polyimide/m-HSNF composites.

Sample	T_5%_ (°C) ^a^	Tg (°C) ^b^	*E*′ (GPa) ^c^	CTE (ppm/°C) ^d^
BPDA-BAPP	521.4	245.9	0.97	59.2
1 wt% BPDA-BAPP/m-HSNF	525.2	248.9	1.16	48.7
3 wt% BPDA-BAPP/m-HSNF	527.3	250.4	1.33	42.3
5 wt% BPDA-BAPP/m-HSNF	528.4	251.4	1.57	38.8
BPDA-HFBAPP	539.7	252.7	0.80	59.1
1 wt% BPDA-HFBAPP/m-HSNF	541.9	255.3	0.97	47.2
3 wt% BPDA-HFBAPP/m-HSNF	543.2	256.5	1.18	44.0
5 wt% BPDA-HFBAPP/m-HSNF	543.8	257.1	1.38	40.9

a: Analyzed by TGA in N_2_ atmosphere b: Analyzed by DMA c: Analyzed by DMA and *E′* is Storage modulus at 50 °C, d: Analyzed by DMA and CTE is calculated at temperature 50~200 °C.

**Table 3 polymers-14-04462-t003:** Tensile strength and elongation at break of polyimide/m-HSNF composites.

Sample	Tensile Strength (MPa)	Elongation at Break (%)
BPDA-BAPP	86.8 ± 1.3	10.5 ± 1.4
1 wt% BPDA-BAPP/m-HSNF	101.9 ± 2.5	9.1 ± 1.9
3 wt% BPDA-BAPP/m-HSNF	109.2 ± 2.7	8.2 ± 1.4
5 wt% BPDA-BAPP/m-HSNF	84.5 ± 3.2	5.9 ± 2.3
BPDA-HFBAPP	82.0 ± 2.4	9.8 ± 0.4
1 wt% BPDA-HFBAPP/m-HSNF	99.1 ± 2.5	9.2 ± 0.9
3 wt% BPDA-HFBAPP/m-HSNF	108.9 ± 3.5	6.7 ± 1.4
5 wt% BPDA-HFBAPP/m-HSNF	84.8 ± 3.3	5.0 ± 0.8

**Table 4 polymers-14-04462-t004:** Dielectric constant of polyimide/m-HSNF composites at 28 and 38 GHz.

m-HSNF Content (wt%)	28 GHz		38 GHz	
	BPDA-BAPP	BPDA-HFBAPP	BPDA-BAPP	BPDA-HFBAPP
0	3.11	2.83	3.09	2.81
1	3.01	2.77	2.98	2.75
3	2.87	2.65	2.85	2.64
5	2.79	2.59	2.76	2.58

**Table 5 polymers-14-04462-t005:** Comparison of PI and PI/m-HSNF developed here and the other sensors reported recently.

Materials	Frequency (GHz)	Dielectric Constant	Dielectric Loss	Ref.
PI	38.9	2.9	0.010	[25]
PI	38.6	3.09	0.010	[26]
BPDA-BAPP	38	3.09	0.0097	This work
5 wt% BPDA-BAPP/m-HSNF	38	2.76	0.0099	This work
BPDA-HFBAPP	38	2.81	0.0088	This work
5 wt% BPDA-HFBAPP/m-HSNF	38	2.58	0.0090	This work

**Table 6 polymers-14-04462-t006:** Transmission loss properties of PI composites at 28 and 38 GHz.

Sample	28 GHz		38 GHz	
	Loss (dB/inch)	Uncertainty	Loss (dB/inch)	Uncertainty
BPDA-BAPP	−2.32	1.4%	−2.80	4.0%
5 wt% BPDA-BAPP/m-HSNF	−2.09	0.2%	−2.40	3.6%
BPDA-HFBAPP	−2.05	1.1%	−2.35	2.9%
5 wt% BPDA-HFBAPP/m-HSNF	−1.90	0.7%	−2.14	3.1%

## Data Availability

The data that support the findings of this study are available on request from the corresponding author.

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
