# Peer review of "Low Dielectric Properties and Transmission Loss of Polyimide/Organically Modified Hollow Silica Nanofiber Composites"

_polymers, 2022, doi:10.3390/polym14204462_

Round 1

Reviewer 1 Report

I reviewed the article with the ID =polymers-1966297.  The article topic is intriguing and promising in the area. Overall, the article structure and content are suitable for the Polymers journal. I am pleased to send you major-level comments, there are some serious flaws that need to be corrected before publication. Please consider these suggestions as listed below. 

·         The title is fine

·         The abstract seems to be fine. Please add one more introductory line of your objective at beginning of the abstract. Highlight the core idea.

·         Keywords are ok but please remove these `` printed circuit board.; ``

·         Research gap should be delivered in a clearer way with the directed necessity for future research work.

·         Introduction section must be written in a more quality way, i.e., more up-to-date references addressed. Please target the specific gap.

·         The novelty of the work must be clearly addressed and discussed, compare previous research with existing research findings, and highlight novelty.

·         Page 1 Line 32 need a single reference, cite this article from literature- Umar, K.; Yaqoob, A.A.; Ibrahim, M.N.M.; Parveen, T.; Safian, M.T. Environmental applications of smart polymer composites. Smart Polym. Nanocompos. Biomed. Environ. Appl. 2020, 15, 295–320. Further remove reference 1-2.

·         The sentence from page 1 Line 33-37 seems very weird, please do not use long sentence. Please revise your paper accordingly since some issue occurs in several spots in the paper. 

·          What is the main challenge? Please highlight this in the introduction part.

·         Page 2-line 67 need also need another reference. Please cite this- Yaqoob, A.A.; Safian, M.T.; Rashid, M.; Parveen, T.; Umar, K.; Ibrahim, M.N.M. Chapter One-Introduction of Smart Polymer Nanocomposites. In Smart Polymer Nanocomposites: Biomedical and Environmental Applications; Elsevier Inc.: Cambridge, MA, USA, 2021; pp. 1–25

·         The main objective of the work must be written in the clearer and more concise way at the end of the introduction section.

·         Please provide space between numbers and units. Please revise your paper accordingly since some issue occurs in several spots in the paper. 

·         Please add the chemical specification and instrumentation brand information.

·          Please check the abbreviations of words throughout the article. All should be consistent.

·         Regarding the replications, authors confirmed that replications of experiment were carried out. However, these results are not shown in the manuscript, how many replicated were carried out by experiment?

·         Please add a comparative profile section to compare your results from previous literature.

·         Please provide high quality figure number 2.

·         Overall result sections seem ok but please follow the journal guidelines to arrange the tables.

·         Section 4 should be renamed by Conclusion and Future perspectives. The conclusion section is missing some perspective related to the future research work, quantifying the main research findings, and highlighting the relevance of the work with respect to the field aspect.

·         To avoid grammar and linguistic mistakes, major level English language should be thoroughly checked. Please revise your paper accordingly since several language issues occur in several spots in the paper.

·         Reference formatting needs careful revision. All must be consistent in one formate. Please follow the journal guidelines.

Author Response

Response to the comments of reviewers

The authors appreciate the referees’ careful reading and thoughtful suggestions. Points by point responses to reviewer’s comments are discussed below.

Reviewer #1:

I reviewed the article with the ID =polymers-1966297.  The article topic is intriguing and promising in the area. Overall, the article structure and content are suitable for the Polymers journal. I am pleased to send you major-level comments, there are some serious flaws that need to be corrected before publication. Please consider these suggestions as listed below.

  1. The title is fine.

Answer: The authors appreciate the reviewer’s comment.

  1. The abstract seems to be fine. Please add one more introductory line of your objective at beginning of the abstract. Highlight the core idea.

Answer: The authors agree with the reviewer’s comment. The authors have added one more introductory line of main objective at beginning of the abstract on page 1, line 9 “In this study, a series of low dielectric constant and transmission loss of polyimide (PI)/organically modified hollow silica nanofibers (m-HSNF) nanocomposites were synthesized via two-step polymerization.” according to the reviewer’s comment.

  1. Keywords are ok but please remove these `` printed circuit board.; ``

Answer: The authors agree with the reviewer’s comment. The “printed circuit board” in keywords have been deleted according to the reviewer’s comment.

  1. Research gap should be delivered in a clearer way with the directed necessity for future research work.

Answer: The authors agree with the reviewer’s comment. The authors have added the text on page 2, line 46 “Most of these measurements are not the 5G high-band frequency.”according to the reviewer’s comment.

  1. Introduction section must be written in a more quality way, i.e., more up-to-date references addressed. Please target the specific gap.

Answer: The authors agree with the reviewer’s comment. The authors have added the text on page 2, line 67 “In order to the investigate the effect of m-HSNF on the circuit transmission loss of PI/m-HSNF nanocomposites, the micro-strip lines fabricated on a series of PI/m-HSNF nanocomposites were used to measure their circuit transmission loss in the 5G high-band frequency.” and page 2 line 75 “Therefore, the dielectric constant of PI/m-HSNF nanocomposites in the 5G high-band frequency was first measured and reported.” according to the reviewer’s comment.

  1. The novelty of the work must be clearly addressed and discussed, compare previous research with existing research findings, and highlight novelty.

Answer: The authors agree with the reviewer’s comment. The authors have added novelty of this research and the comparison with existing research findings in Table 5 according to the reviewer’s comment.

  1. Page 1 Line 32 need a single reference, cite this article from literature- Umar, K.; Yaqoob, A.A.; Ibrahim, M.N.M.; Parveen, T.; Safian, M.T. Environmental applications of smart polymer composites. Smart Polym. Nanocompos. Biomed. Environ. Appl. 2020, 15, 295–320. Further remove reference 1-2.

Answer: The authors agree with the reviewer’s comment. The authors have added the reference “Umar, K.; Yaqoob, A.A.; Ibrahim, M.N.M.; Parveen, T.; Safian, M.T. Environmental applications of smart polymer composites. Smart Polym. Nanocompos. Biomed. Environ. Appl. 2020, 15, 295–320.” as reference 1 of this manuscript and have deleted original reference 1-2 according to the reviewer’s comment.

  1. The sentence from page 1 Line 33-37 seems very weird, please do not use long sentence. Please revise your paper accordingly since some issue occurs in several spots in the paper.

Answer: The authors agree with the reviewer’s comment. The authors have shortened the sentence on page 1, line 33-37 according to the reviewer’s comment.

  1. What is the main challenge? Please highlight this in the introduction part.

Answer: The authors agree with the reviewer’s comment. The authors have added the main challenge in the introduction part according to the reviewer’s comment.

  1. Page 2-line 67 need also need another reference. Please cite this- Yaqoob, A.A.; Safian, M.T.; Rashid, M.; Parveen, T.; Umar, K.; Ibrahim, M.N.M. Chapter One-Introduction of Smart Polymer Nanocomposites. In Smart Polymer Nanocomposites: Biomedical and Environmental Applications; Elsevier Inc.: Cambridge, MA, USA, 2021; pp. 1–25

Answer: The authors agree with the reviewer’s comment. The authors have added the reference “Yaqoob, A.A.; Safian, M.T.; Rashid, M.; Parveen, T.; Umar, K.; Ibrahim, M.N.M. Chapter One-Introduction of Smart Polymer Nanocomposites. In Smart Polymer Nanocomposites: Biomedical and Environmental Applications; Elsevier Inc.: Cambridge, MA, USA, 2021; pp. 1–25” as reference 18 of this manuscript according to the reviewer’s comment.

  1. The main objective of the work must be written in the clearer and more concise way at the end of the introduction section.

Answer: The authors agree with the reviewer’s comment. The authors have rewritten the text in the manuscript on page 2, line 75 “Therefore, the dielectric constant of PI/m-HSNF nanocomposites in the 5G high-band frequency was first measured and reported.” according to the reviewer’s comment.

  1. Please provide space between numbers and units. Please revise your paper accordingly since some issue occurs in several spots in the paper. 

Answer: The authors agree with the reviewer’s comment. The authors have modified the text in the manuscript according to the reviewer’s comment.

  1. Please add the chemical specification and instrumentation brand information.

Answer: The authors agree with the reviewer’s comment. The authors have added the chemical formula and purity as well as instrumentation brand information according to the reviewer’s comment.

  1. Please check the abbreviations of words throughout the article. All should be consistent.

Answer: The authors agree with the reviewer’s comment. The authors have checked the abbreviations of words throughout the article according to the reviewer’s comment.

  1. Regarding the replications, authors confirmed that replications of experiment were carried out. However, these results are not shown in the manuscript, how many replicated were carried out by experiment?

Answer: The authors agree with the reviewer’s comment. The authors have added the number of experiments on page 3, line 135 “The average values of experimental data shown here are obtained from at least three measurements.” according to the reviewer’s comment.

  1. Please add a comparative profile section to compare your results from previous literature.

Answer: The authors agree with the reviewer’s comment. The authors have added the comparison between this study and previous reports in Table 5 according to the reviewer’s comment.

  1. Please provide high quality figure number 2.

Answer: The authors agree with the reviewer’s comment. The high-quality figure 2 has been used according to the reviewer’s comment.

  1. Overall result sections seem ok but please follow the journal guidelines to arrange the tables.

Answer: The authors agree with the reviewer’s comment. The authors have followed the journal guidelines to arrange the tables according to the reviewer’s comment.

  1. Section 4 should be renamed by Conclusion and Future perspectives. The conclusion section is missing some perspective related to the future research work, quantifying the main research findings, and highlighting the relevance of the work with respect to the field aspect.

Answer: The authors agree with the reviewer’s comment. The authors have renamed the Section 4 according to the reviewer’s comment.

  1. To avoid grammar and linguistic mistakes, major level English language should be thoroughly checked. Please revise your paper accordingly since several language issues occur in several spots in the paper.

Answer: The authors have checked and modified the grammar mistakes according to the reviewer’s comment.

  1. Reference formatting needs careful revision. All must be consistent in one format. Please follow the journal guidelines.

Answer: The authors agree with the reviewer’s comment. The authors have checked and modified the reference formatting according to the reviewer’s comment.

Reviewer 2 Report

Manuscript ID: polymers-1966297 entitled:

Low dielectric properties and transmission loss of polyimide/organically modified hollow silica nanofiber composites

Authors

Shu-Yang Lin , Yu-Min Ye , Erh-Ching Chen , Tzong-Ming Wu

General comment

The study presents the results obtained regarding both the synthesis of two different polyimides using two types of diamine monomers, with or without fluorine-containing groups and biphenylene dianhydride structure, and some nanocomposites based on PI/organically modified silica hollow nanofiber as well as their testing as a substrate for use in 5G technology printed circuit board.

Some recommendations and observation remain:

1. Insert the notations and composition of the prepared and analyzed samples in the experimental part. Mention in text the number of replicates for each determinations.

2. Additional information from APTEOS is needed to clarify the syntheses, the bibliographic citation mentioned by authors cannot be accessed [18= Chien HC, Lin SY, Chen EC, Wu TM Synthesis and dielectric properties of polyimide/hollow404 silica nanofiber composite, J Mater Sci - Mater Electron (submitted). The composition of composite is missing.

3. Compare the present results with others already published in the literature on composites suitable for 5G printed circuit board.

4.  It is good to revise the whole text and, where possible, use short sentences. Rephrase the sentences for better understanding. Some examples:

A R 73 From these EDS images, the elements of carbon, and silicon are appropriately separated and distributed in the composite samples.”

At 185 “This occurrence is assigned to the incorporation of m-HSNF contained higher thermal stability in the polyimide matrix, causing the thermal stability improvement of the composites.”

At R240 “The improvement of tensile strength is contributed to the incorporation of the inorganic and stiff m-HSNF, resulting in the reinforcement effect to improve the rigidity of the BPDA-BAPP.”

At R250 “Two different kinds of substance existed, such as polyimide matrix and m-HSNF, when current went through the polyimide nanocomposites via the dielectric test.”

5. Since water has a relatively high dielectric constant, and the dielectric constant of the composite will increase with water absorption, it is recommended to present data on moisture absorption and adhesion to a substrate of the prepared composites.

6. It is well known that fluorinated polyimides with various structures have been exploited to reduce the dielectric constant of the systems. It is necessary to be emphasized more the novelty of the research in the introduction part, and in conclusions insert the obtained results in a more concise manner.

Author Response

Response to the comments of reviewers

The authors appreciate the referees’ careful reading and thoughtful suggestions. Points by point responses to reviewer’s comments are discussed below.

Reviewer #2:

The study presents the results obtained regarding both the synthesis of two different polyimides using two types of diamine monomers, with or without fluorine-containing groups and biphenylene dianhydride structure, and some nanocomposites based on PI/organically modified silica hollow nanofiber as well as their testing as a substrate for use in 5G technology printed circuit board. Some recommendations and observation remain:

  1. Insert the notations and composition of the prepared and analyzed samples in the experimental part. Mention in text the number of replicates for each determinations.

Answer: The authors agree with the reviewer’s comment. The authors have added the notations and composition of the prepared samples in the experimental part on page 3 line 106 “After the mixture was completely dissolved, the BPDA with the same mole of BAPP or HFBAPP was added and stirred for 12 h under the condition of ice-water bath to form a homogeneous polyamic acid (PAA) of BPDA-BAPP (C43H32N2O8) or BPDA-HFBAPP (C43H26F6N2O8). The obtained PAA solution was poured onto the clean glass plate and heated in an oven at 80 °C to eliminate the solvent. Then the thermal imidization of PAA was performed at 100 °C/1 h, 150° C/1 h, 200 °C/1 h, 250 °C/1 h, 300 °C/1 h in vacuum to form polyimide (PI) of BPDA-BAPP (C43H30N2O7) or BPDA-HFBAPP (C43H24F6N2O7).” according to the reviewer’s comment. The authors have also added the number of experiments on page 3 line 135 according to the reviewer’s comment.

  1. Additional information from APTEOS is needed to clarify the syntheses, the bibliographic citation mentioned by authors cannot be accessed [18= Chien HC, Lin SY, Chen EC, Wu TM Synthesis and dielectric properties of polyimide/hollow404 silica nanofiber composite, J Mater Sci - Mater Electron (submitted). The composition of composite is missing.

Answer: The authors agree with the reviewer’s comment. The authors have added the preparation of m-HSNF using APTEOS on page 3, line 99 “The fabricated HSNF surfaces were functionalized by C9H23NO3Si (APTEOS) coupling agent by dispersing 1g HSNF in 200 mL ethanol and then 2 mL APTEOS dispersed in 35 mL ammonium hydroxide was added in HSNF solution in a three-necked flask while vigorously stirred at room temperature under ultra-sonication for 32 h to obtain APTEOS-modified HSNF (m-HSNF).” according to the reviewer’s comment.

  1. Compare the present results with others already published in the literature on composites suitable for 5G printed circuit board.

Answer: The authors agree with the reviewer’s comment. The authors have found only two references measured the dielectric properties in the 5G high-band frequency (> 24 GHz) and have added the results in Table 5 according to the reviewer’s comment.

  1. It is good to revise the whole text and, where possible, use short sentences. Rephrase the sentences for better understanding. Some examples:

A R 73 From these EDS images, the elements of carbon, and silicon are appropriately separated and distributed in the composite samples.”

At 185 “This occurrence is assigned to the incorporation of m-HSNF contained higher thermal stability in the polyimide matrix, causing the thermal stability improvement of the composites.”

At R240 “The improvement of tensile strength is contributed to the incorporation of the inorganic and stiff m-HSNF, resulting in the reinforcement effect to improve the rigidity of the BPDA-BAPP.”

At R250 “Two different kinds of substance existed, such as polyimide matrix and m-HSNF, when current went through the polyimide nanocomposites via the dielectric test.”

Answer: The authors agree with the reviewer’s comment. The authors have modified the whole text according to the reviewer’s comment.

  1. Since water has a relatively high dielectric constant, and the dielectric constant of the composite will increase with water absorption, it is recommended to present data on moisture absorption and adhesion to a substrate of the prepared composites.

Answer: The authors agree with the reviewer’s comment. The authors have added the contact angle and water absorption of polyimide/m-HSNF composites in the supporting information according to the reviewer’s comment. It is clear that the contact angle of composites was slightly increased with the addition of m-HSNF , but the water absorption was almost the same with the addition of m-HSNF. The data of contact angle and water absorption for the polyimide/m-HSNF composites are also listed below.

Table . Contact angle and water absorption of polyimide/m-HSNF composites.

Sample

Contact angle (degree)

Water absorption (%)

BPDA-BAPP

68.3

0.97

1 wt% BPDA-BAPP/m-HSNF

71.8

0.96

3 wt% BPDA-BAPP/m-HSNF

73.3

0.96

5 wt% BPDA-BAPP/m-HSNF

74.2

0.96

BPDA-HFBAPP

73.7

0.74

1 wt% BPDA-HFBAPP/m-HSNF

74.8

0.74

3 wt% BPDA-HFBAPP/m-HSNF

77.5

0.73

5 wt% BPDA-HFBAPP/m-HSNF

79.9

0.72

  1. It is well known that fluorinated polyimides with various structures have been exploited to reduce the dielectric constant of the systems. It is necessary to be emphasized more the novelty of the research in the introduction part, and in conclusions insert the obtained results in a more concise manner.

Answer: The authors agree with the reviewer’s comment. The fluorinated polyimides have been exploited to reduce the dielectric constant. In this study, the incorporation with m-HSNF can further reduced the dielectric constant of fabricated materials in both PI system. The authors have added the novelty of this research in the introduction part and discuss the obtained results in conclusion according to the reviewer’s comment.

Round 2

Reviewer 1 Report

Accepted in the present form

Author Response

The authors appreciate the reviewer’s comment.

Reviewer 2 Report

Manuscript ID: polymers-1966297 entitled:

Low dielectric properties and transmission loss of polyimide/organically modified hollow silica nanofiber composites

Authors

Shu-Yang Lin , Yu-Min Ye , Erh-Ching Chen , Tzong-Ming Wu

General comment

The study presents the results obtained regarding both the synthesis of two different polyimides using two types of diamine monomers, with or without fluorine-containing groups and biphenylene dianhydride structure, and some nanocomposites based on PI/organically modified silica hollow nanofiber as well as their testing as a substrate for use in 5G technology printed circuit board.

Some recommendations and observation remain:

1. Additional data must be mentioned in the text.

Insert in text the explanation write in Response to the comments of reviewers:  “The data of contact angle and water absorption for the polyimide/m-HSNF composites are also listed in Supplementary data Table S1.” It is clear that the contact angle of composites was slightly increased with the addition of m-HSNF, but the water absorption was almost the same with the addition of m-HSNF.

Number the table in the in supplementary data.

Author Response

Response to the comments of reviewers

The authors appreciate the referees’ careful reading and thoughtful suggestions. Points by point responses to reviewer’s comments are discussed below.

Reviewer #2:

The study presents the results obtained regarding both the synthesis of two different polyimides using two types of diamine monomers, with or without fluorine-containing groups and biphenylene dianhydride structure, and some nanocomposites based on PI/organically modified silica hollow nanofiber as well as their testing as a substrate for use in 5G technology printed circuit board. Some recommendations and observation remain:

  1. Additional data must be mentioned in the text. Insert in text the explanation write in Response to the comments of reviewers:  “The data of contact angle and water absorption for the polyimide/m-HSNF composites are also listed in Supplementary data Table S1.” It is clear that the contact angle of composites was slightly increased with the addition of m-HSNF, but the water absorption was almost the same with the addition of m-HSNF. Number the table in the in supplementary data.

Answer: The authors agree with the reviewer’s comment. The authors have added the discussion of Supplementary data Table S1 on page 11, line 291 “Since water has a relatively high dielectric constant, the data of contact angle and water absorption for the polyimide/m-HSNF nanocomposites are also listed in Supplementary data Table S1. It is clear that the contact angle of composites was slightly increased with the addition of m-HSNF, but the water absorption was almost the same with the addition of m-HSNF. Therefore, the effect of water absorption on the dielectric constant of fabricated polyimide/m-HSNF nanocomposites can be ignored.“ according to the reviewer’s comment.
